# *Cutibacterium acnes*: An Emerging Prostate Cancer Pathogen

**DOI:** 10.3390/biology15010030

**Published:** 2025-12-24

**Authors:** Luka Brajdic, Ella K. Reed, Helen B. Pearson, Helen L. Brown

**Affiliations:** 1School of Biosciences, Cardiff University, Sir Martin Evans Building, Cardiff CF10 3AX, UK; 2The European Cancer Stem Cell Research Institute, Division of Biomedicine, School of Biosciences, Cardiff University, Cardiff CF24 4HQ, UK; reedek@cardiff.ac.uk (E.K.R.);

**Keywords:** *Cutibacterium acnes*, prostate cancer, microbiome

## Abstract

*Cutibacterium acnes* is a bacterium which normally lives on the skin, and can contribute to the development of acne. Over the last 20 years, *C. acnes* has also been found in the prostate of people with prostate cancer. This review discusses research aiming to understand how often this bacterium is found in the prostate, how it survives there, and what effects it may have on prostate health. Studies show that *C. acnes* can be found in early-stage and advanced prostate tumours, as well as some healthy prostates. The bacteria may be able to survive in the prostate because of the low oxygen levels and acidic surroundings. Within this environment, *C. acnes* can trigger inflammation and change the body’s immune response. Evidence also suggests that the bacterium can contribute to DNA damage in healthy prostate cells, change how cells use energy, and affect the structure surrounding the cells. In these ways, *C. acnes* may influence the development of cancer. However, there are disagreements in study methods because *C. acnes* can also contaminate biological samples. This makes it difficult to understand whether the bacterium contributes to cancer or is found by coincidence. Using new genomic tools may help to tell the difference between the two in future research. Signs of *C. acnes* have also been found in urine, the gut, and in the blood, which suggests that it might be used to improve early detection of the disease. More research is needed to understand exactly what *C. acnes* is doing in prostate cancer, and whether there is a way to exploit it and help improve care for people with prostate cancer.

## 1. Introduction

In 2022, breast, prostate, and lung cancer were reported to have the highest global incidence per 100,000 people, posing a significant health risk and financial hurdle to global populations [1]. Prostate cancer (PCa) is the most common male urogenital cancer, with 1.4 million new cases reported in 2020 predicted to rise to 2.9 million by 2040 [2]. While many men diagnosed with PCa now have a good prognosis, some patient cohorts are at higher risk of mortality, including those that present with advanced disease at diagnosis, have a family history of PCa, or are from a high-risk ethnic group (e.g., sub-Saharan African heritage) [1,3]. Early detection has been shown to significantly improve patient outcome, especially in high-risk populations, but existing biomarkers lack specificity [4,5]. Serum prostate-specific antigen (PSA) levels are widely used to screen for PCa. However, as PSA is produced by malignant and normal/hyperplastic prostate luminal epithelial cells, the PSA test can lead to false positives (i.e., an elevated PSA level), owing to an inability to distinguish between malignant prostate tumours and benign prostate hyperplasia (BPH). BPH is a relatively common non-malignant condition where the normal prostate becomes enlarged and produces more PSA, reported to occur in up to 50% of men over the age of 50, with symptoms often resembling those of PCa [4,6]. Accordingly, researchers have spent considerable effort exploring new avenues for diagnostic and prognostic markers for PCa [7]. There have been multiple recent and ongoing trials to assess the suitability of urinary and tissue genomics-based markers [7,8,9], and there is an increasing interest in exploiting markers from the human microbiome as well as understanding its role in PCa pathogenesis [10,11,12,13].

This review aims to identify gaps in the understanding of the role of *Cutibacterium acnes* infection during PCa, with a particular focus on the mechanisms that underpin *C. acnes* persistence and tumorigenicity. Through the collation of current knowledge in the field and critical evaluation of emerging evidence exploring the mechanisms underpinning *C. acnes* colonisation and tumorigenesis, we will explore the importance of this bacterial opportunistic pathogen as a carcinogen, as well as its potential diagnostic and predictive biomarker value.

### 1.1. Prostate Cancer Progression

Structurally, the prostate comprises three zones: central, transition, and peripheral, with the peripheral zone being the origin of most prostate tumours [14]. The most common histological form of PCa is prostate adenocarcinoma, although neuroendocrine (NE) differentiation, sarcomas and lymphomas may also arise [15]. The prostatic stroma consists of fibroblasts, smooth muscle, and immune-infiltrating cells. Androgen receptor (AR)-expressing luminal epithelial cells secrete prostatic fluid and PSA, while AR-low/negative basal cells and AR-negative NE cells provide structural and functional support. PCa can originate from both luminal and basal cells [16], with alterations in DNA damage response (DDR) genes, such as mutation of the tumour suppressor gene Breast Cancer Gene 2 (*BRCA2*), being a truncal event in PCa carcinogenesis [15,17]. Malignant transformation then typically progresses through precursor lesions of prostatic intraepithelial neoplasia, where high-grade cells expressing markers associated with adenocarcinoma may be found [18,19].

Tumour aggressiveness is clinically assessed using the Gleason grading system, endorsed by the World Health Organization. Each abnormality observed during biopsy is graded from 1 (tissue resembling normal prostate) to 5 (highly undifferentiated tissue) [15]. The Gleason score is calculated by adding the two most prevalent patterns, reported as a single number (e.g., 7) or as two numbers (e.g., 3 + 4 or 4 + 3). Scores of six or higher are further classified into five grade groups to improve prognostic stratification. Patterns above three are considered malignant, and men with Gleason scores ≥ 6 typically require treatment or close monitoring. While widely used, the Gleason system has limitations, including risks of over- or under-treatment [15].

Early detection of PCa remains a cornerstone for improving patient outcomes. Localised PCa has a 10-year survival rate approaching 99%, whereas advanced disease with distant metastases reduces survival to approximately 30% at five years [20]. This stark contrast underscores the need for timely diagnosis and intervention. Advances in genomics, imaging, and microbiome-based biomarkers offer promising avenues to refine risk stratification and personalise screening strategies.

### 1.2. Microbial Pathogens and Cancer Development

Microbial pathogens are increasingly recognised as pivotal players in cancer biology. Chronic infection can promote tumorigenesis through sustained inflammation, genomic instability, and immune modulation [21]. *Helicobacter pylori* is a well-established bacterial carcinogen, classified as a Group I carcinogen owing to its strong association with gastric cancer [22,23]. Similarly, *Fusobacterium nucleatum* has been implicated in colorectal cancer progression, while oral pathogens such as *Porphyromonas gingivalis* correlate with oesophageal cancer risk [24,25] Beyond bacteria, oncogenic viruses including Epstein–Barr virus, human papillomavirus, and hepatitis B and C viruses have also been shown to drive malignancy through integration into host genomes and disruption of cell cycle regulation [26]. Mechanistically, pathogens can induce DNA damage, alter epigenetic landscapes, and manipulate signalling pathways such as NF-κB and STAT3, fostering a microenvironment conducive to neoplastic transformation. Emerging evidence also highlights the role of microbiome dysbiosis in modulating hormonal and immune responses, influencing cancer susceptibility and therapeutic outcomes [27].

### 1.3. Therapeutic Use of Pathogens in Oncology

Beyond their pathogenic roles, microbes and their derivatives are being repurposed as innovative tools in cancer therapy. Attenuated or genetically engineered bacteria, such as *Listeria monocytogenes* and *Salmonella* spp., have been explored as vectors for delivering tumour antigens and stimulating robust immune responses [28,29]. Protozoan parasites, notably *Toxoplasma gondii* and *Trypanosoma cruzi*, exhibit remarkable immunomodulatory properties that can reverse tumour-associated immunosuppression [30,31] Non-replicating *T. gondii* strains have demonstrated efficacy in preclinical models of melanoma, ovarian, and pancreatic cancer by activating Th1 immunity, enhancing Interleukin (IL)-12 and Interferon Gamma (IFN-γ) production, and promoting CD8+ T-cell infiltration [32] Similarly, molecules secreted by *T. cruzi*, such as calreticulin and P21, exert antiangiogenic effects, inhibit tumour cell migration, and induce apoptosis, while mucin-like antigens trigger cross-reactive immune responses against tumour-associated glycans [33]. These strategies exploit the evolutionary capacity of pathogens to engage innate and adaptive immunity, offering a blueprint for next-generation immunotherapies. Coupled with advances in nanotechnology for targeted antigen delivery [34], pathogen-based interventions hold promise for overcoming resistance and improving outcomes in refractory cancers.

## 2. Microbiome and Prostate Cancer

Microbiomes are ecosystems of resident bacteria, viruses and fungi that play a significant role in maintaining human health or causing disease [35]. Emerging research demonstrates that microbes can be carcinogenic, immunomodulatory and can even impact treatment efficacy [36]. Indeed, the most recent update to the hallmarks of cancer acknowledged the role of microbes in enabling tumorigenesis, given that the polymorphic variability of individuals’ microbiomes causes profound differences in cancer risk and progression [27,37]. For example, chronic infection with *Helicobacter pylori* significantly increases gastric cancer risk, leading to the organism being classified as a group I carcinogen [22,23]. Similarly, colonisation of the gastrointestinal (GI) tract by *F. nucleatum* is linked to development and progression of colorectal cancer [24], while *Porphyromonas gingivalis* has been associated with oesophageal cancer [25].

More recently, the interface between bacterial members of the human microbiome and PCa has come under the spotlight. The microbiomes of the gut, urinary tract, and the local tumour tissue have all been associated with PCa (Figure 1) [10,12,38]. Indirect pathways have been proposed to link dysbiosis of the gut microbiome to PCa [39]. Laaraj et al. (2025) review the crosstalk between gut microbiota and PCa, which links fatty acid metabolism and microbial-derived androgens to disease aggressiveness and tumour growth [38]. The authors describe how the proposed relationship between PCa and the gut microbiome is complex and can be both pro-tumorigenic and anti-tumorigenic. Laaraj et al. (2025) also note that PCa is now recognised to alter the composition of the gut microbiome, indicating bidirectional influence between the two organs [38]. In particular, the authors discuss the multifaceted role of bacterially derived short-chain fatty acids (SCFAs), which can both promote and constrain local inflammation and gut permeability. Increased gut permeability is proposed as a potential mechanism enabling bacterial translocation from the gastrointestinal tract to the prostate. The authors further report that several gut-resident bacterial species are not only affected by host hormone levels but are also capable of metabolising hormones such as testosterone and its precursor dihydrotestosterone, both of which are important in PCa progression [38]. The production of SCFA by *C. acnes* is well established, in particular its production of propionic acid [40]. *C. acnes*-produced SCFA have been shown to have both a proinflammatory effect on sebocytes [41], and inhibit the persistence of *Staphylococcus epidermidis* [42], allowing *C. acnes* to compete with this fast-growing species when in the same niche.

### 2.1. Role of the Urinary Microbiome in Prostate Cancer

The urinary microbiome is another emerging key player in prostatic diseases. There appears to be differences in urinary microbiota composition between patients with chronic prostatitis, BPH, and PCa. In a systematic review of the literature, Mjaess et al. (2023) identified in their systematic review that *Pseudomonas*, a genus common in the normal urinary microbiota of men, was often absent from PCa urinary microbiomes, while *Lactobacillales* was overrepresented in men with BPH but not elevated in PCa or prostatitis [43]. In contrast, *Streptococcus* and *Escherichia* were commonly shared genera across all three pathologies [43]. Recently, Hurst and colleagues identified anaerobic bacteria from the genera *Fenollaria*, *Peptoniphilus*, *Anaerococcus*, *Porphyromonas*, and *Fusobacterium* in urine sediment, urine extracellular vesicles (EVs) (discussed in detail in Section 4.4), and prostatic tissue of men with PCa [10]. The presence of these microbes was significantly associated with poor progression-free survival and patients with worse prognosis could be successfully identified by the composition of their urinary microbiota. As well as the presence of specific microbial species, the likelihood of microorganisms being present within the urine is reported to correlate with increased cancer severity [10]. Taken together, these results suggest bacterial biomarkers present promising diagnostic/prognostic tools for clinical application and a novel route for therapeutic intervention. Nevertheless, whether there is a causative relationship between these bacteria and PCa and the extent of the underlying molecular mechanisms involved remain unclear.

### 2.2. Prostatic Tissue Microbiome and Prognostic Significance

Although the composition of solid tumour tissue microbiome communities has historically been disputed in the literature [44], it is now generally accepted that solid tumours harbour unique and distinctive tissue microbiomes [13,45]. Several studies have identified a diverse range of bacteria, which may exist within both healthy prostate tissue and PCa. Cavarretta et al. (2017) analysed the distribution of bacterial load in radical prostatectomy samples across tumour, peri-tumour, and non-tumour tissue [46]. *Cutibacterium* and *Corynebacterium* genera were the most abundant in all sample types, while *Staphylococcus* was significantly more frequent in tumour and peri-tumour tissue compared to non-tumour tissue [46]. However, the translatability of results was limited by the study’s sample size of only 16 men with PCa, and its lack of inclusion of benign control samples.

Subsequently, a larger cohort of 65 men with treatment-naive PCa, who had undergone radical prostatectomy, was studied by Feng and colleagues [47]. Microbes from the genera *Escherichia*, *Acinetobacter*, *Pseudomonas*, and *Propionibacterium* (the latter genus includes *Cutibacterium* species) were identified as core members of the prostatic tissue microbiome, abundant in the metagenome and metatranscriptome of both cancerous and matched-normal tissue [47]. Of note, infection of the prostate by uropathogenic *Escherichia coli*, along with *Enterococci* subspecies, is also a common cause of chronic prostatitis [48], which has been proposed as a risk factor for the development of PCa [39]. Analysis of clinical data for all 65 specimens revealed no significant difference in the microbial community composition between malignant and adjacent benign tissue, nor a relationship with Gleason score, the system used to clinically grade primary PCa tumours [47].

Conversely, Kim et al. (2024) [49] reported bacterial composition differs depending on the Gleason grade in a small collection of radical prostatectomy samples from 26 patients with treatment-naive PCa. High-grade prostate tumours (defined as Gleason grade 3–5) contained *Cutibacterium*, *Pelomonas*, and *Corynebacterium* genera at higher frequency than low-grade prostate tumours (defined as Gleason grade 1 or 2). A correlative relationship between the prostatic tissue microbiome and PCa progression was also suggested by Banerjee et al. (2019), who conducted an array-based metagenomic and capture-sequencing study on tissue specimens from 50 malignant prostatectomy samples and 15 BPH specimens, which included bacterial and non-bacterial members of the microbiome [13]. The authors reported a diverse variety of bacterial species within the PCa microbiome including *Cutibacterium acnes*, *Chlamydia trachomatis*, and *H*. *pylori* (>90% of cases) that were at least a 2-fold increase in PCa samples compared to BPH. Interestingly, hierarchical cluster analysis of microbiome signatures from the bacteria, fungi, and viruses found in PCa samples revealed three different groups, suggesting the existence of multiple distinct PCa microbiomes [13]. Further analysis revealed correlations of certain average microbiome hybridisation signatures with different grades and stages of disease; for example, signatures from *Helicobacter*, human papilloma virus 18, Kaposi-Sarcoma-associated Herpesvirus, and members of the polyomaviridae family were higher in low-grade PCa (Gleason score < 6) while signatures from the parasitic roundworm genus *Trichinella* was associated with high-grade PCa (Gleason scores 8 and 9). Together, these data support the existence of rich and diverse microbiomes within PCa tumours compared to benign tissue and suggest a potential relationship with disease aggressiveness. However, future studies with larger patient cohorts are needed to provide robust data with sufficient statistical power, and additional investigations are needed to understand how the presence of intratumoral bacteria may contribute to disease initiation, progression, or treatment response.

## 3. Prevalence of *Cutibacterium acnes* in Tumour Microbiomes

Arguably the most frequently reported species within the prostate microbiome is *Cutibacterium acnes* (previously termed *Propionibacterium acnes*) [12,46,47,49,50,51]. *C. acnes* is a dominant member of the human microbiome, primarily residing in the sebaceous glands of the skin. In low abundance, it has also been detected in the microbiome of the oral cavity, gastrointestinal, and urogenital tracts [52,53,54]. As a normal member of the skin flora, *C. acnes* contributes to the maintenance of homeostasis and restricts microbiome colonisation of harmful bacterial species [55,56]. *C. acnes* is also an opportunistic pathogen, playing a role in acne vulgaris, as well as being identified in implant-related infections, endocarditis and sarcoidosis [54,57,58]. Here, we explore the prevalence of *C. acnes* in cancer, particularly PCa, and discuss how *C. acnes* can support and thrive within the prostate tumour microenvironment.

### 3.1. Occurrence of Cutibacterium acnes Across Cancer Types

*C. acnes* was reported to colonise prostate tumours for the first time in 2005 and has been identified in multiple studies over the following two decades [12,50,51,59,60]. The microbe can be distinguished into three major subspecies; *C. acnes* subsp. *acnes* (type I), *C. acnes* subsp. *defendens* (type II), and *C. acnes* subsp. *elongatum* (type III); they can be further grouped by six corresponding phylotypes (IA1, IA2, IB, IC, II, and III) (Figure 2A). This discrimination is based on multi-locus sequence typing, rDNA, hly hemolysin, and recA DNA repair genes [61]. In addition to providing insight into the genomic diversity of the species, this system also allows the association of phylotypes with different disease states. For example, a loss of *C. acnes* phylotype diversity and enrichment of type IA1 strains is thought to drive acne development—type IA1 *C. acnes* are generally considered the most immunogenic, and immunogenicity has been observed to decrease in mixed phylotype skin models in vitro [62,63]. Interestingly, type IB and II *C. acnes* have been associated with both healthy skin but also implant-associated infections and prostate cancer, suggesting that host-beneficial or -detrimental interaction may depend on the microenvironment [54,60,64,65]. Beyond PCa, *C. acnes* has recently also been reported in several other malignancies including gastric cancer [66], pancreatic cancer [67], thyroid cancer [68], bladder cancer [69], and ovarian cancer [70] (Figure 2B).

While *C. acnes* has been frequently reported within PCa samples, the relationship between the two is complicated by highly variable detection rates that range from 23% to 95% incidence [71]. Similarly, *C. acnes* has also been identified within non-PCa prostate tissue [47,51,65]. Indeed, due to the frequency of *C. acnes* in negative control samples [64,72] some studies chose to preclude the reporting of *C. acnes* from their results; Banerjee et al. (2013) excluded *Propionibacterium* signatures from the prostate specific microbiome due to its detection in tissue-free control samples [13].

The wide detection range can, in part, be explained by methodological differences in detection and sampling, and it is possible that at least some of the *C. acnes* detected within biopsy samples can be attributed to contamination from the skin during sampling. Indeed, *C. acnes* was detected in 20 human tissue types via polymerase-chain-reaction-based methods; however, only in three tissue types (basal cell carcinoma, malignant melanoma, and breast cancer) were relative *C. acnes* levels higher than 1 ppm [73]. Since *C. acnes* is found in high abundance on sebaceous skin, such as the groin, it was suggested that during the collection of punch biopsies *C. acnes* would be transferred from its natural home within the skin’s sebaceous glands into the prostate tissue. This argument was also made by some to explain the presence of the bacterium on prosthetic joint implants, which were considered to have failed following “aseptic loosening” [74]. *C. acnes* is now accepted as an atypically presenting true infectious agent within implant infections [75,76], requiring fastidious collection and culture conditions to be successfully identified. However, its abundance on the skin and difficulty to eradicate using standard skin decontamination procedures [77] means that studies attempting to detect *C. acnes* within the prostate must carefully consider their sampling strategy and collection protocols to minimise the risk of sample contamination with *C. acnes*. While the risk of sample contamination is not negligible, the identification of *C. acnes* intracellularly within prostate cells [12], clearly shows that *C. acnes* is colonising and persisting within prostate tissue rather than being introduced during sample collection.

### 3.2. Tumour Microenvironment and Cutibacterium acnes Persistence

Theoretically, the prostate tumour microenvironment provides good conditions for *C. acnes* colonisation with regard to oxygenation and acidity. The prostate can be divided into three structural zones: central, transition, and peripheral, the latter being the origin of most prostate tumours [14]. Prostate tumours themselves are structurally heterogeneous, meaning that multiple physiological niches are likely to be present both within each tumour and across the whole prostate. Hypoxia is common in solid tumours, but has been described as ‘profound’ in some prostate and pancreatic tumours [78]. Oxygen levels in the normal prostate are comparatively low at a median of 4%, and untreated prostate tumours are reported to contain over 12× less oxygen than the healthy prostate [79]. While *C. acnes* can survive in the presence of oxygen [80], it is generally considered to be an anaerobic species [54], and as such, would be able to survive and proliferate in hypoxic conditions associated with PCa.

*C. acnes* also thrives in the acidic skin environment, where it contributes to maintaining low pH (approx. pH 5) through degradation of sebum triglycerides via secretion of lipases, and production of the short-chain fatty acid propionic acid as part of its normal metabolism [54,81]. While tumour pH is difficult to determine in situ, the interstitial space of solid tumours is reportedly more acidic than healthy tissue [82,83]. Tumour acidity is closely associated with metabolic reprogramming, a hallmark of cancer, due to the cellular shift to prefer energy generation through glycolysis even under aerobic conditions, termed the Warburg effect [21,84]. Intracellular persistence of bacteria has also been observed to induce similar metabolic shifts [85], so it is plausible to think that *C. acnes* may not only benefit from the prostate tumour niche, but help maintain it.

## 4. Pathogenic Mechanisms of *Cutibacterium acnes* in Prostate Cancer

While multiple studies have now identified *C. acnes* as a member of the PCa tumour microbiome [12,47,51,86], it remains to be determined whether *C. acnes* is acting as a commensal organism that has found a favourable niche, and if the presence of *C. acnes* is actively driving disease onset, progression and/or mediating treatment response. Although the host–pathogen relationship with *C. acnes* is best characterised within the common skin condition acne [87], advances over recent years have described mechanisms by which *C. acnes* may contribute to carcinogenesis [12,51,59,70,88]. Here, we review the four key molecular mechanisms whereby *C. acnes* persistence could facilitate prostate tumorigenesis and progression (Figure 3); (a) increased inflammation, immune cell activation, and macrophage infiltration, (b) DNA damage and inhibition of repair mechanisms, (c) modification of extracellular matrix components and (d) secretion of bacterial extracellular vesicles containing virulence factors enabling *C. acnes* persistence.

### 4.1. Inflammatory and Immune Effects of Cutibacterium acnes

Over the last decade, multiple researchers have shown that *C. acnes*, like many other bacterial species, can stimulate an inflammatory response by the host [51,89,90,91]. While intraprostatic inflammation is common in PCa, PCa is generally considered to be an immunologically cold cancer, where the presence and activity of immune cells is low [92,93]. As such, the presence of *C. acnes*, which can stimulate inflammation when detected by the host, alters the tumour microenvironment [51,94]. *C. acnes* has been identified to survive intracellularly in several cell types, including epithelial cells [12,86,95], keratinocytes [95], macrophages [59,96], mesenchymal stem cells [97], osteoblasts and osteoclasts [97]. Thus, it has been speculated that intracellular persistence of *C. acnes* may aid the organism’s ability to evade the immune system, allowing chronic persistence and inflammation [96] to facilitate tumorigenesis [98].

Bae et al. (2014) reported an increased presence of stromal macrophages in both malignant and healthy prostate tissue from radical prostatectomy when *C. acnes* was present, and that the number of stromal macrophages infected with *C. acnes* positively correlates with acute and chronic inflammation [51]. Similarly, Cohen et al. (2005) identified an association of *C. acnes* infection in the prostate with inflammation [50]. Shinohara et al. (2013) demonstrated that inoculating wild-type C57BL/6J mice with a prostatectomy-derived isolate of *Cutibacterium acnes* via urethral catheterisation induced inflammation in the murine prostate in vivo [94]. The infection triggered both acute and chronic inflammatory responses, accompanied by increased Ki-67-positive epithelial cells and reduced Nkx3.1 expression, a marker of proliferative inflammatory atrophy in humans which is AR controlled [94]. These findings provide evidence that *C. acnes* can initiate prostatic inflammation.

Macrophage activation by *C. acnes* has been observed in PCa, leading to an increase in cytokine production including tumour necrosis factor alpha (TNF-α) and IFN-γ [51] (Figure 3a). Mechanistically, synthesis of type I interferons in human macrophages can be induced by *C. acnes* infection through the cGAS-STING pathway that triggers an inflammatory response upon the detection of stress-induced cytosolic DNA [90]. More recently, Davidsson et al. (2021) described that *C. acnes* infection of PCa cells can upregulate immunosuppressive genes in macrophages, including immune checkpoint receptor PD1 ligand (PD-L1) and CC motif chemokine ligand-17 and -18 (CCL17 and CCL18) [59]. Retrospective analysis of regulatory T-cells (Treg) cell infiltration also revealed a positive association of Treg cells in tumour stroma and epithelia with the presence of *C. acnes* [59].

As noted earlier, *C. acnes* is not just detected within PCa (Figure 3). Emerging evidence suggests that *C. acnes* also exhibits similar pro-inflammatory effects to those found in the prostate within different cancer types, indicating a common mechanism. In an in vivo model of ovarian cancer, *C. acnes* infection by intratumoral injection led to an upregulation of proinflammatory cytokines IL-1-beta (IL-1β) and TNF-α and activation of the Hedgehog signalling pathway, which can modulate the tumour immune microenvironment through driving tumour-associated macrophage proliferation, T-cell activation, and upregulation of PD-L1 [70]. Li et al. (2021) reported that conditioned media (CM) from *C. acnes-infected* macrophages significantly increased the migration of gastric cancer cells in vitro, while *C. acnes* infection alone did not [66]. Furthermore, macrophages infected with *C. acnes* produced CM, which contained increased levels of IL-6, IL-10, Chemokine Receptor type 2 (CCR-2), and macrophage inflammatory protein (MIP)-1β. IL-10 and CCR-2 are markers of M2-like macrophages, which are considered pro-tumorigenic due to their immunosuppressive properties [59,66], and it is reported that in vitro infection of the monocyte cell line THP-1 with *C. acnes* can induce M2-like macrophage polarisation through toll-like receptor (TLR)-4/PI3K/Akt signalling [66]. Together, these findings support a role for *C. acnes* in driving disease progression through the modulation of macrophage activity and the establishment of a pro-tumour immune microenvironment, although additional in vivo work is required to fully establish this link.

### 4.2. Impact on DNA Repair and Metabolism

In addition to its immunomodulatory activity, additional tumorigenic mechanisms driven by *C. acnes* are emerging in the literature, including attenuation of the DNA damage repair pathway and metabolic reprogramming. For instance, Ashida et al. (2024), recently proposed a carcinogenic role for *C. acnes* in PCa through downregulation of homologous recombination repair (HRR) and Fanconi anaemia pathways in normal prostate cells (Figure 3b) [12]. The authors observed that host cell invasion was associated with cancer-associated cellular changes, including downregulation of the tumour suppressor genes *BRCA2* and *RAD51* [12]. Furthermore, the results demonstrated the proportionate induction of double-strand break marker γH2AX foci relative to the quantity of *C. acnes*, increasing significantly in infected cells compared to those cultured with heat-inactivated *C. acnes*. This suggests that *C. acnes* infection leads to DNA double stranded breaks and HRR deficiency, a known initiator of malignant transformation [99,100]. However, infection did not impact *BRCA2* protein expression and no methylation in the *BRCA2* promoter region was observed, thus it is currently unclear whether this activity has a functional impact in PCa [12] and additional work is warranted.

Deregulation of metabolic programming is a well-known hallmark of cancer [21]. Infection by intracellular bacteria can shift host-cell metabolism to prefer aerobic glycolysis, resembling the Warburg effect that can promote cancerous growth [85]. As discussed in Section 3.1, *C. acnes* can persist in multiple cell types [12,96], it is therefore plausible that *C. acnes* can modulate the cellular metabolism of multiple cell types within the tissue it infects, instigating a metabolic switch that favours cancer growth. Indeed, bacteria capable of intracellular persistence, for example *Mycobacterium tuberculosis*, can infect and replicate within macrophages, leading to increased glycolysis and accumulation of lipid bodies which creates a suitable niche for its persistence [101]. Furthermore, Almoughrabie et al. (2023), reported increased relative abundance of lipids (including free fatty acids, triglycerides, ceramides, and cholesterol) in normal keratinocytes in response to *C. acnes* conditioned media, which occurred across different *C. acnes* phylotypes [102], suggesting a similar phenomenon may occur in other epithelial tissues. Previous research has demonstrated an association between increased fatty acid uptake in PCa and lipid biomass, as well as oncogenic lipid signalling using patient-derived PCa tissue and xenograft mouse models [103]. Blocking the fatty acid transporter CD36-mediated fatty acid uptake led to a reduction in cancer growth in xenograft models, suggesting it as an avenue to enhance treatment efficacy in PCa [103]. Moreover, Almoughrabie et al. (2023) described *C. acnes*-induced accumulation of glycerol-3-phosphate-acyltransferase (GPAT)-3, mediated by the propionic acid within the *C. acnes* conditioned media [102]. In hepatocellular carcinoma, GPAT3 can contribute to carcinogenesis and treatment resistance by inhibiting apoptosis through the reprogramming of triglyceride metabolism [104]. Its mitochondrial isoform, GPAT2, is highly expressed in PCa and promotes growth and tumorigenicity of breast cancer cells in vitro, but it is unclear whether GPAT2 responds to *C. acnes* like its isoform [105,106]. In summary, data to date suggest that *C. acnes* may promote PCa tumorigenicity through dysregulation of DNA repair pathways as well as increasing host cell lipid synthesis.

### 4.3. Extracellular Matrix Modification by Cutibacterium acnes

Remodelling of the extracellular matrix (ECM), which is composed of glycosaminoglycans, collagens, fibronectin, elastin and hyaluronic acid (HA) plays a key role in mediating cancer invasiveness, epithelial-to-mesenchymal transition (EMT) and treatment resistance [107]. Proteins contained in the *C. acnes* secretome include glycoside hydrolases, esterases, proteases, Christie–Atkins–Munch-Petersen (CAMP) factors, and glyceraldehyde 3-phosphate dehydrogenase (GAPDH) [108]. Of these, several are capable of ECM modification (Figure 3c)—particularly the release of glycoside hydrolases, β-N-acetylglucosaminidase and muramidase lytic enzymes and metalloproteases, lipases, proteases, and hyaluronidase (HAase) by *C. acnes* that reportedly cause disruption of the follicular epithelium in acne [109]. Additionally, Nazipi et al. (2017) reported that type IB/II strains of *C. acnes*, two commonly detected strains within PCa samples [54,60,64,65], contained a variant of hyaluronate lyase (HYL-IB/II), which was shown to contribute to HA degradation [110]. This variant may therefore contribute to the increased invasiveness of type IB/II *C. acnes* strains that was observed in vitro using a keratinocyte–sebocyte co-culture model by Spittaels et al. (2020), and perhaps explaining its prevalence in deep tissue infections and PCa compared to other *C. acnes* subtypes [64,111].

In addition to secreting ECM-modifying components, *C. acnes* is also capable of stimulating the release of ECM-modifying proteins from infected host cells. Inflammatory acne lesions, which have been reported to contain an overabundance of *C. acnes* [112], are associated with increased activation of matrix metalloproteinase (MMP) expression and collagen degradation compared to skin without acne [113]. Mechanistically, *C. acnes* is reported to directly stimulate host cell production of TNF-α [114], which in turn modulates MMP activity and subsequent collagen degradation in the dermis [113]. In support of *C. acnes* coordinating the release of ECM remodelling factors from its host cells, Fassi Fehri et al. (2011) studied the response of normal prostate epithelial cells (RWPE1) to a prostatectomy-derived type IB isolate of *C. acnes* using transcriptional profiling, revealing 425 deregulated genes, 49% of which were cancer-associated, and several implicated in ECM remodelling [86]. The ECM remodelling genes identified including four plasminogen activation system genes (*uPA*, *uPAR*, *PAI-1*, *PAI-2*), which are significant contributors to ECM substrate degradation and MMP activation [115]. *C. acnes* infection, as well as *C. acnes* CM also increased the host cell expression of MMPs and secretion of functional MMP-9 in vitro [86]. Importantly, MMP-9 expression is associated with high-grade PCa tumours [116], biochemical recurrence [117], and advanced stages of disease [118] as well as ECM remodelling [119].

Secretion and stimulation of ECM remodelling factors promoted by *C. acnes* can have host-beneficial or -detrimental effects. Increased mechanical ECM stiffness, facilitated largely by collagen crosslinking density and HA deposition, is associated with more invasive phenotypes and can confer treatment resistance in advanced disease [120,121]. The degradation of HA and collagens in the presence of *C. acnes* may therefore soften tumour tissue and stroma by reducing collagen crosslinking density and HA deposition, potentially exhibiting anti-tumorigenic effects. However, because intracellular *C. acnes* can increase host cell expression of MMPs in prostate cells [86] that help facilitate invasiveness [119], it also has the potential to accelerate disease progression. In addition, a softer matrix in PCa tumours has been demonstrated to promote AR-mediated gene expression, which is crucial during early stages of disease and facilitates advanced PCa [122]. It is therefore plausible to think that the adaptation of the PCa-associated type IB/II *C. acnes* to HA degradation, as well as its implication in MMP activation and the plasminogen system could impact PCa disease initiation and invasiveness through ECM modification. Accordingly, future work should address not just the effect of *C. acnes* on epithelial and immune cell function, but consider the impact on the tumour microenvironment at large, including stromal cells, ECM components, interaction with other members of the tumour microbiome, and the contribution this may play in metastatic disease.

### 4.4. Tumorigenic Effects of Secreted Products and Vesicles

Bacterial extracellular vesicles (bEVs) are an emerging field of interest owing to their potential as biomarkers and drug delivery systems. However, they are also a major mechanism by which virulence factors can be delivered into host cells. Previous work with the gut bacterium *Akkermansia muciniphila* demonstrated the feasibility of bEVs in the circulation entering the prostate [123]. Upon *A. muciniphila* intravenous injection into an immune-competent syngeneic tumour mouse model of PCa, bEVs were traced to the tumour site, leading to a reduction in tumour burden and recruitment of M1-like macrophages via CD8+ T cell signalling [123]. While in this instance the presence of bEVs was protective, bEVs have been demonstrated to contain pathogen-associated molecular pattern molecules which resemble those of the parent bacteria [124]. To the best of our knowledge, no studies have yet been performed that explore the effect of *C. acnes* bEVs on PCa. However, studies in allied areas suggest that *C. acnes* bEVs will likely play a key role in permitting *C. acnes* to colonise the prostate, triggering inflammation and driving malignant changes (Figure 3d). Notably, Jeon et al. (2017) identified 252 vesicular proteins in *C. acnes* bEVs, many of which are relevant to its pathogenicity, including adherence, virulence, and immunogenicity factors [125]. As discussed in Section 3.1, the presence of *C. acnes* within the prostate is a significant driver of inflammation [86] and studies exploring the role of *C. acnes* in driving inflammation in acne also note the proinflammatory effects *C. acnes*-derived bEVs [62,91], further supporting the notion that *C. acnes* can elicit a proinflammatory response within prostate tissue. Future work addressing this directly would be of great interest to the field and could provide both a greater understanding of *C. acnes* PCa pathogenicity but may also provide future biomarker targets.

Remarkably, Jingushi et al. (2024), reported enhanced *C. acnes* DNA in serum-isolated bEVs in renal cell carcinoma patients, while *C. acnes* bEVs promoted renal carcinoma cell proliferation in vitro [126]. Importantly, within a subcutaneous renal cell carcinoma xenograft model, administration of *C. acnes* via an intraperitoneal injection also increased the number of Ki-67-positive renal cell carcinoma cells, as well as the number of CD31-positive endothelial cells, indicating that *C. acnes* can potentially regulate angiogenesis [126]. Bacterial EVs have also previously been shown to translocate from the gut to distant organs via the circulatory system [127]. The systemic circulation of *C. acnes* bEVs (and indeed other bacterial and host EVs) makes them an attractive diagnostic tool, easily detectable in body fluids such as serum, saliva and urine [128].

## 5. Therapeutic and Diagnostic Potential of *Cutibacterium acnes*

Beyond the tissue microbiome, *C. acnes* has also been identified in the urinary microbiome of PCa patients [10], and previous work has shown that *C. acnes* is significantly more abundant in the urine of bladder cancer patients compared to healthy volunteers [69]. Thus *C. acnes* may prove to hold predictive value for a range of cancers. In keeping with this, a relationship between commensal members of the gut microbiome and patients’ response to oncologic interventions, particularly immunotherapies, has also been identified [129,130]. The abundance of *C. acnes* in the gut microbiome has been associated with treatment response in melanoma, non-small cell lung cancer (NSCLC) and nasopharyngeal carcinoma (NPC) [131,132]. For instance, Diop et al. (2025), found *C. acnes* was significantly enriched in faecal specimens collected from patients with advanced melanoma and NSCLC that were resistant to immune checkpoint inhibitor therapy relative to those that responded [131]. Similarly, *C. acnes* infection in tumour tissue and faecal samples was among the top three key discriminators between NPC patients whose tumours were radiotherapy responsive or non-responsive, with *C. acnes* relative abundance elevated in the responsive group compared to non-responders [132]. While these data emphasise the potential diagnostic power of *C. acnes* in human malignancies, it is clear that more research is needed to truly harness its potential value. This is particularly important for prostate cancer, where a study did not find any correlation between *C. acnes* infection in PCa and advanced disease [10].

### Potential Role in Prostate Cancer Treatment

The immunomodulatory properties of *C. acnes* have also been explored therapeutically. The first publication in this area reported reticuloendothelial stimulation following *C. acnes* injection, at the time named *Corynebacterium parvum* [133]. Following injection, activation of macrophage activity [134,135] was observed, making *C. acnes* a target of interest as a vaccine adjuvant [136,137]. A recent study reported that immunisation of BALB/c mice with a DNA vaccine target alongside *C. acnes* achieved a higher rate of immunity, increased IFN-γ production, and recognition of vaccine-encoded peptides [138]. The observed immunomodulatory effects which make *C. acnes* suitable as a vaccine adjuvant have been attributed to secretion of interferon and proinflammatory cytokines, activation of TLR-2 and TLR-9, and myeloid differentiation primary response 88 (MyD88) receptor activation, as well as enhancement of the Th1 population function [137].

During the 1960s, researchers also discovered that heat- and phenol-killed suspensions of *C. acnes* with anti-tumorigenic effects, which were further explored in murine models [134,139,140]. Although extensive research suggested a promising anti-tumorigenic role for *C. acnes*, toxicity and reduced survival associated with attenuated *C. acnes* has stalled its clinical use. Adverse effects were recorded in a clinical trial using *C. acnes* as postoperative adjuvant immunotherapy of stage I and II NSCLC, where intrapleural *C. acnes* administration led to a significantly higher rate of side effects such as fever and chest pain [141].

More recent work has also indicated an antitumour effect for *C. acnes*, and its close relative *Cutibacterium granulosum* for the treatment of melanoma. Tsuda et al. (2011), showed that injection of heat-killed *C. acnes* from healthy volunteers reduced tumour size in female C57BL/6J (B6) mice injected with B16 melanoma cells [142]. Histopathological analysis revealed granuloma formation at the tumour site when *C. acnes* was present, alongside elevated expression of the pro-inflammatory markers IFN-γ and TNF-α [142], thus harnessing *C. acnes* ability to trigger a strong immune response. A further metagenomic investigation of the tumour-associated microbiome of biopsies originating from 47 cutaneous melanoma patients indicated that high abundance of both *C. acnes* and *C. granulosum* was associated with better overall survival and lower recurrence [143]. When the same analysis was carried out for biopsy samples from nasal and oral melanoma (N = 12 for each sample type), a clear correlation could not be determined for *C. acnes*, and *C. granulosum* was negatively correlated to tumour recurrence in these samples [143]. This study highlights that further work using larger patient cohorts is needed to understand why different tumours, and perhaps even individual patients, respond so differently to the presence of *C. acnes*.

## 6. Future Prospects

### 6.1. Beyond Biopsy Contamination

Once considered as simply a biopsy contaminant, *C. acnes* is now acknowledged as a frequent resident within prostate tissues, albeit not detected in all PCa cases (Table 1). Variability in detection rates across studies is reflective of both methodological disparities between studies and our incomplete understanding of *C. acnes* diversity and behaviour within the prostate niche. The bacterium has been associated with premalignant lesions [12], but also with locally advanced disease [49]. This inconsistency highlights the need for improved and consistent study design with sufficient statistical power, as well as deeper analysis of the strain and genomic resolution of the bacterial isolates identified. While it is now generally accepted that the majority of *C. acnes* detected within the PCa biopsies is not simply due to transfer of the organism from the patient’s skin, there is still little understanding of how prostate colonisation by *C. acnes* is spatially related to tumour site, PCa grade or clinical outcome. We propose these questions are the focus of future research efforts to understand the role of *C. acnes* in PCa.

### 6.2. Adaptation Within the Prostate Niche

PCa develops over a long time period, often moving along a spectrum from benign inflammatory disease to localised malignancy and finally, if not detected and treated earlier, invasive and ultimately treatment-resistant disease [148]. As such, by the time of diagnosis and samples becoming available to researchers and clinicians, it is likely that a significant period of time has passed in which multiple microbial species, including *C. acnes*, have invaded and colonised the tumour site, interacting with both the host and other microbes present to form malignancy-associated communities. Therefore, the *C. acnes* which is detected and characterised during PCa sampling may have little in common phenotypically, and perhaps genotypically, with the initial colonising isolate. It is well established that within other chronic diseases, bacterial species undergo significant niche adaptation to be able to successfully persist. Within cystic fibrosis (CF), a well-studied disease associated with chronic bacterial colonisation of the lung [149], research has shown that *Pseudomonas aeruginosa*, a bacterial pathogen which is a major cause of morbidity and mortality in people with CF, undergoes significant evolution within the CF lung [150]. This evolutionary adaptation often leads to slower growth, altered metabolism, increased biofilm formation, and greater tolerance of antibiotics [150]. Interestingly, analysis has shown that multiple strains of *P. aeruginosa* can colonise an individual’s lung, each existing within individual niches [151]. This suggests that, alongside isolate evolution, it is likely that at least some people with CF experience multiple *P. aeruginosa* colonisation episodes. *C. acnes* is also well adapted to persist within low-oxygen environments throughout the human body and its implication in multiple different implant-related infections is testament to the adaptability of the species [152]. Similar patterns of colonisation and evolution are very likely to take place within the prostate; however, at the time of writing, this remains an unexplored research area. As noted within Section 4.1, colonisation of prostate tissue by *C. acnes* results in chronic inflammation [50,51]. It is widely recognised that chronic inflammation cannot only drive the initial development of malignancies, but also contributes significantly to their progression [153]. It should therefore be assumed that, at the very least, the presence of *C. acnes* could contribute to driving malignant changes within the prostate through increased inflammation. However, as the research discussed in Section 4 clearly indicates, *C. acnes* is not merely contributing passively to tumour development via inflammatory pathways but instead is actively modulating the prostate environment to support its persistence, a consequence of which is to drive malignant changes. As noted in Section 2, the local PCa environment is colonised by multiple different bacterial species. The sebaceous gland, where *C. acnes* typically resides, also contains a rich and diverse microbiome [57,154] with *C. acnes* able to produce multiple compounds allowing it to effectively compete within this environment [155]. The presence of other microbes within the environment is likely to influence the behaviour, proteome and secretome of *C. acnes*, and so it may be that not just the presence of *C. acnes*, but its combination with specific other microbes, is an important factor in driving the tumorigenic potential of both *C. acnes* and the wider prostate microbial community.

### 6.3. Challenges in Accessing Prostate Cancer Samples

The detailed picture of colonisation, adaptation, and competition within the CF lung was only possible because researchers could carry out longitudinal sampling of the same cohorts. This is easier for CF than PCa due to the accessibility of the sample site, as sputum is the most common CF lung sample type. It can be readily provided by those with CF and is routinely collected for diagnostic and surveillance purposes [156]. The bacterial isolates identified within sputum are routinely cultured, banked for long-term storage, and genetically and phenotypically characterised [156], which provides researchers with a wealth of information reflecting both changes over time and within specific lung sites.

For patients with PCa, tissue or bacterial samples are typically taken at diagnosis, with late-stage disease biopsies and longitudinal sampling being less common, partly reflecting the inaccessible nature of the prostate [157], but also the fact that the primary disease is typically removed during first-line therapy. Unfortunately, this means that a significant body of information about both the very early (BPH-PCa transition) and late stages (including metastatic disease) of prostate microbial colonisation is lost. Similarly, the prostate is a complex organ with multiple lobes, often containing multiple heterogeneous malignancies. Even when multiple biopsies are collected, the whole prostate will not be represented, and some malignant areas may be missed [158]. This may also be the reason why previous researchers have reported such a wide detection range for *C. acnes*-positive PCa samples (as discussed in Section 3).

### 6.4. Unresolved Mechanistic Questions

Several recent advances suggest that *C. acnes* may exert multiple tumour-promoting effects (Figure 3). These include modulation of host immune responses [59], promotion of chronic inflammation [50,51], interference with DNA repair pathways [12], host cell metabolism reprogramming [102,104], and ECM remodelling [86,122]. Moreover, the detection of *C. acnes* genomic material beyond tumour tissue, such as the urinary and gut microbiomes, as well as circulatory bEVs, opens avenues for non-invasive diagnostics [10,126,132]. Still, it remains unclear whether *C. acnes* acts as a driver of oncogenesis, a co-factor in disease progression, or a bystander thriving in a permissive niche. Advancing this field demands robust in vitro and in vivo models capable of capturing the interplay between microbial, tumour, and host systemic factors.

### 6.5. Translational Potential of Cutibacterium acnes Extracellular Vesicles

The potential of *C. acnes*-derived EVs as both a future diagnostic marker, active driver of proinflammatory and cancerous changes and potentially a future therapeutic target is very significant, and we highlight these questions as a particularly promising area of future research. Research effort is already underway to explore the diagnostic and mechanistic potential of host-derived EVs in human malignancies (for a recent review, see Tang et al. (2025) [159]) and interest in now also increasing in bacterial equivalents and their relevance to various cancers [160]. The potential of *C. acnes* EVs to drive malignant changes within the prostate is discussed in Section 4.4, and speculations from the small number of studies available suggest that *C. acnes*-derived EVs may potentially be utilised as a diagnostic marker [10].

Interestingly, the proinflammatory properties of *C. acnes* bEVs [91] and whole cells [142] also indicate that *C. acnes* may present a novel PCa therapeutic target, potentially enhancing the efficacy of immunotherapies. Interest in bacterial-based cancer therapies is increasing (for a recent review see Tieu et al. (2024) [161]), with the field encompassing both the use of bacterial-derived components (such as EVs and peptides) as well as whole bacteria (both genetically modified and unmodified) to infiltrate and disrupt the tumour microenvironment and deliver payloads which stimulate the host immune system. Unfortunately, reports of adverse effects within human trials [141] indicate that, as with many of the other questions around the role of *C. acnes* within PCa, further research is required to fully understand how *C. acnes* might be used safely to support future PCa treatment strategies.

## 7. Conclusions

*C. acnes* has emerged as a frequent and functionally relevant member of the PCa microbiome. This review synthesises evidence from two decades of research, highlighting its potential to influence PCa pathogenesis through immune modulation, DNA damage, metabolic reprogramming, and extracellular matrix remodelling. The presence of *C. acnes* in urinary and gut microbiomes, and detection of *C. acnes* derived extracellular vesicles in the circulation, suggests diagnostic potential beyond the tumour site. However, detection rates vary widely across studies, reflecting methodological differences and the need for improved sampling and microbial community- and bacterial strain-level resolution.

Importantly, *C. acnes* appears capable of adapting to the prostate tumour microenvironment, persisting intracellularly and interacting with host cells in ways that may promote malignancy. Its ability to influence immune cell behaviour, stimulate inflammation, and alter host metabolism and tissue architecture positions it as a candidate driver of disease progression. Future research should prioritise longitudinal studies, spatial mapping of microbial colonisation, and functional characterisation of bacterial extracellular vesicles. Understanding how *C. acnes* interacts with other microbial species and host factors may also reveal new insights into tumour ecology and identify novel biomarkers or therapeutic targets. Clarifying whether *C. acnes* acts as a bystander, biomarker, or active driver of PCa remains a key challenge for the field.

## Figures and Tables

**Figure 1 biology-15-00030-f001:**
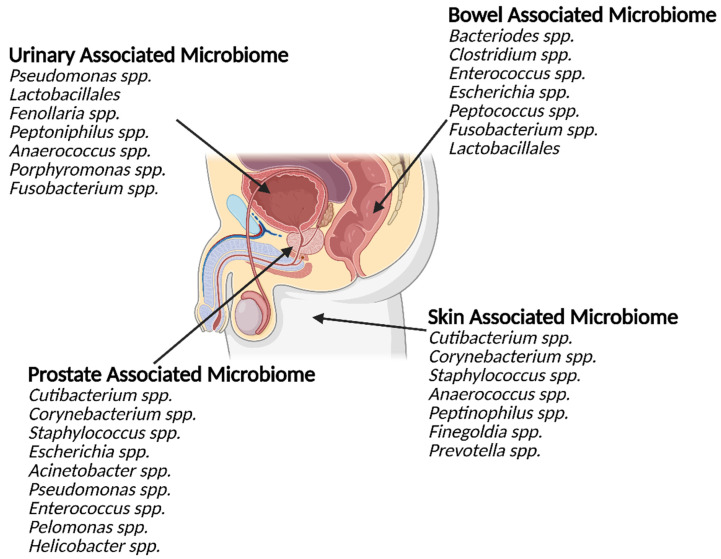
The microbiomes of the urinary tract, gastrointestinal tract, prostate and groin skin share similarities. The prostate is in close proximity to several large microbiome communities (external and internal). These communities share significant overlap in the genus they contain. The bacterial genus commonly detected within the prostate, both in health and disease, also shares similarities with the other local microbiome communities.

**Figure 2 biology-15-00030-f002:**
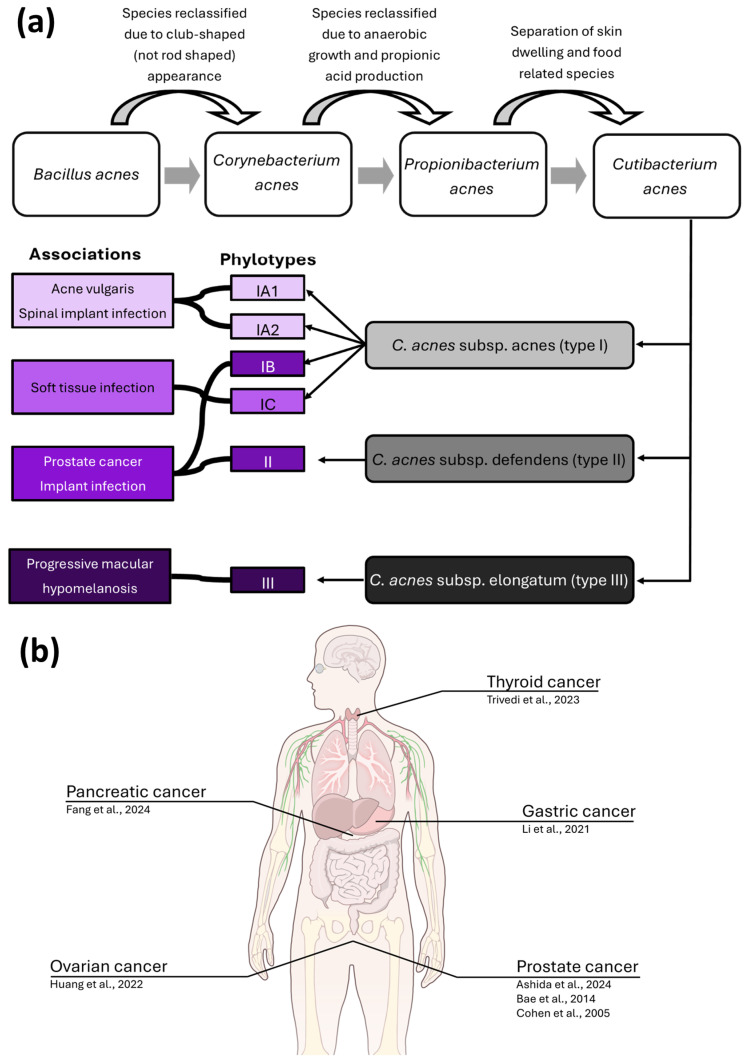
*C. acnes* is associated with multiple diseases and has been identified in several cancers. (**a**) Diagrammatic representation of the name changes and classification system associated with *C. acnes*. Each of the *C. acnes* subspecies is associated with one or more phylotypes, which are associated with several pathological conditions. While phylotypes are overrepresented within these pathologies, they are not exclusive to them, and each phylotype is detected in a range of human niches, both within health and disease. (**b**) cancer types (with author affiliations) *C. acnes* has been identified in [12,50,51,66,67,68,70]. Illustration from NIAID Visual & Medical Arts. (7 October 2024). Human Anatomy. Illustration created using NIAID NIH BIOART Source: https://bioart.niaid.nih.gov/bioart/519 (accessed on 1 December 2025).

**Figure 3 biology-15-00030-f003:**
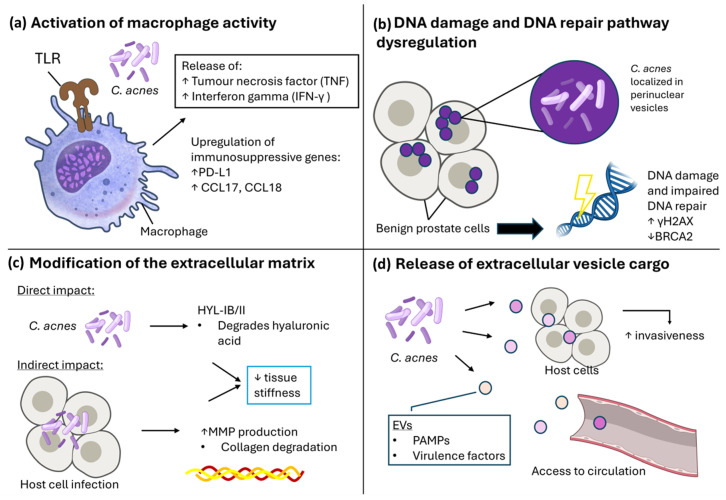
*C. acnes* can exhibit pro/tumorigenic effects on host cells through multiple mechanisms. (**a**) Activation of macrophage activity. *C. acnes* can stimulate macrophages through toll-like receptors (TLR), leading to release of tumour necrosis factor (TNF) and Interferon gamma (IFN-γ), as well as upregulation of immunosuppressive genes PD-L1 and CCL17/18. (**b**) DNA damage and DNA repair pathway dysregulation. When localised in perinuclear vesicles, intracellular presence of *C. acnes* was associated with DNA damage as indicated by increased γH2AX, as well as impaired DNA repair pathway regulation through downregulation of BRCA2. (**c**) Modification of the extracellular matrix. Direct changes can be exhibited by *C. acnes* through its hyaluronate lyase activity, which degrades hyaluronic acid. Indirectly, *C. acnes* can also lead to increased host cell MMP production during infection, which can lead to collagen degradation and subsequent decrease in ECM tissue stiffness. (**d**) Release of extracellular vesicle cargo. *C. acnes* release EVs containing PAMPs and virulence factors, which can be internalised by host cells to lead to increased invasiveness, or access circulation. Illustration created using NIAID NIH BIOART. Source: https://bioart.niaid.nih.gov/ (accessed on 30 September 2025).

**Table 1 biology-15-00030-t001:** Studies detected *C. acnes* in prostate tissue across the last 20 years ordered by recency.

Source	Location	Methods	Main Findings	Limitations
Ashida et al., 2024 [12]	Japan	Radical prostatectomy samples from 20 patients. Candidate samples derived by RNA-seq with de-novo assembly of non-human reads, PCR, and immunohistochemistry.	*C. acnes* among four main pathogen candidates. Identified in 15/20 samples from RNA-seq and 19/20 samples by immunohistochemistry. Predominantly identified in prostatic intraepithelial neoplasia (PIN).	No PCR results for *C. acnes* due to lack of optimisation.
Bidaud et al., 2020 [144]	France	Samples isolated using transrectal needle biopsy (‘biopsy sterile gun’). Colonies identified using MALDI-TOF mass spectrometry, followed by molecular typing of *C. acnes* strains detected	*C. acnes* observed in 5.6% (2/36) of samples belonging to phylotypes IA1, IB, and II. Prostate adenocarcinoma patient (*n* = 1) had phylotype IA1 usually on skin. *Cutibacterium avidum* also found in three patients.	Contamination risk from needle biopsy.
Feng et al., 2019 [47]	China	A total of 65 radical prostatectomy samples and matched normal tissue analysed by shotgun integrated metagenomic and meta transcriptomic analysis.	*C. acnes* identified among most abundant genera in prostatic tissue. No difference in alpha diversity between benign and malignant tissue microbiota, including *C. acnes*, or between different tumour stages.	Matched normal tissue used as control.
Ugge et al., 2018 [145]	Sweden	Pre-operative biopsy samples were taken from PC patients who later underwent radical prostatectomy. Serum levels of IL6 and CXCL8 (pro-inflammatory mediators) in blood from patients prior to surgery determined by ELISA.	*C. acnes* identified in 60 out of 99 patient samples. *C. acnes* associated with PCa in case-control setting, like in previous studies. *C. acnes* induced IL6 and CXCL8 secretion by prostate epithelial cells. No statistical difference in serum IL6 and CXCL8 levels between samples with or without *C. acnes* infection. Authors conclude that inflammation caused by *C. acnes* may be low grade with no impact on systemic IL6 and CXCL8 levels.	Samples used stem from pre-operative biopsy rather than radical prostatectomy. Authors critique themselves the lack of CRP as measure of systemic inflammation and wider panel of pro- and anti-inflammatory markers needed.
Yow et al., 2017 [146]	Australia	Two approaches to massively parallel sequencing (MPS) using 20 snap-frozen prostate tissue cores from “aggressive” (as defined by Gleason score and TNM stage) PCa cases (removed by radical prostatectomy).	Identified *Enterobacteriaceae* common in all samples. Other operational taxonomic unit (OTU)s in 95% of samples included *P. acnes* among *Enterobacteriaceae* and *Streptococcaceae*, *Staphylococcus*, *Escherichia*, *Moraxella*, *Propionibacterium acnes* and *Streptococcus pseudopneumoniae.* These OTUs contribute large proportion of relative abundance of total community sequences across 20 samples. Relative contribution of each of the seven was relatively consistent across samples.	Lack of control tissue, just PCa tissue.
Davidsson et al., 2016 [65]	Sweden	Radical prostatectomy samples from 100 men with PCa and 50 controls (bladder cancer patients). Cultures were grown from six biopsies each for seven days. Each isolate was sequenced to the species level. Subsequent infection cell work to assess acute and chronic presence.	*C. acnes* more common in prostate carcinoma than controls (however some still grew in controls). Multivariable analysis—men with *C. acnes* have a 4-fold risk of prostate cancer diagnosis adjusted for age, year of surgery and smoking status. In cellular co-culture with prostate cell line PNT1A an increase in proliferation and cytokine/chemokine secretion in infected cells was recorded.	Controls are not healthy—men with bladder cancer.
Bae et al., 2014 [51]	Japan	Bacterial screen of radical prostatectomy samples from 28 prostate cancer patients and 18 bladder cancer patients using enzyme immunohistochemistry with *C. acnes* specific monoclonal antibody (PAL) and NF-kB antibody.	Immunohistochemistry of PAL Ab showed small, round bodies in non-cancerous glandular epithelium and stromal macrophages. PC samples had higher freq. cytoplasmic *C. acnes* or nuclear NF-kB expression of glandular epithelium. Increased number of stromal macrophages with *C. acnes* and nuclear NF-kB more frequent in glands with *C. acnes*. Number of stromal macrophages with *C. acnes* correlated with grade of inflammation. Suggesting that intraepithelial *C. acnes* infection in non-cancerous prostate glands as well as inflammation caused by *C. acnes* may contribute to tumorigenesis. “	Controls are not healthy—men with bladder cancer.
Fassi et al., 2011 [86]	Germany	10 swabs taken from central zone of prostate specimens after needle biopsy were identified using in situ immunofluorescence (ISIF). Isolates were then co-cultured with PCa cell line RWPE1, after which microarray and transcriptome analysis were performed.	*C. acnes* was found in 81.7% of PC samples (58/71), and none of the healthy controls (n = 20) and other malignant tissue (mamma carcinoma, n = 59). No correlation between presence of *C. acnes* and Gleason score. Infection of prostate cells resulted in inflammatory host cell response, inc. secretion of IL-6 and IL-8, likely mediated by activated transcriptional factors NF-kB and STAT3. Longer term exposure impacted cell proliferation and growth, associated with cellular transformation.	Needle biopsy—contamination risk.
Sfanos et al., 2008 [147]	USA	Tested 170 samples of prostate tissue core as well as seminal vesicles (SV) from 30 cancer patients by universal eubacterial PCR, looking for 16S rDNA gene sequences. Positive PCR products were then cloned and sequenced. In addition, samples from 30 patients were cultured microbiologically, as well as using DNA samples from another 200 patients to test by organism-specific PCR including *C. acnes*.	A total of 83 distinct microorganisms were found by 16S sequencing, while microbiological culture isolated significantly fewer species. No sig. association between any species and histologic indication of acute or chronic inflammation. *C. acnes* found in 8/200 (4%) of prostate core and 2 SV samples.	Analysis using 16S rDNA sequencing could have picked up on microbial DNA from engulfment by phagocytic immune cells like macrophages.
Alexeyev et al., 2006 [72]	Sweden	Using archival prostate samples from 352 patients with BPH, common microbes were identified and evaluated differences between those who later went on to develop prostate cancer. 16S RNA.	In 27% of samples, 16S RNA was detected. *C. acnes* was the most common organism at 23% of RNA-pos samples. A total of 62% of these samples exhibited severe histological inflammation vs. 50% in bacteria-negative samples. Presence of *C. acnes* associated with development of prostate cancer.	Archival samples—tissue preservation may have impacted microbial communities. Prostate samples were acquired through transurethral resection of prostate (TURP), so exposure to contaminants is viable via this route.

## Data Availability

Not applicable.

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
