# Peer review of "Cutibacterium acnes: An Emerging Prostate Cancer Pathogen"

_biology, 2025, doi:10.3390/biology15010030_

Round 1
Reviewer 1 Report
Comments and Suggestions for Authors
I have attached a PDF file.

Author Response
Reviewer 1 comments
This review discusses the potential role of Cutibacterium acnes in prostate cancer. The authors have included several important and relevant studies suggesting that C. acnes may contribute to prostate cancer development and progression. However, in several sections, the manuscript shifts attention toward the involvement of C. acnes in other malignancies rather than maintaining focus on prostate cancer. To strengthen the manuscript's central theme, the authors should empathize with prostate cancer-specific evidence and clearly distinguish comparative examples from other cancers. The review would benefit from careful revision to improve clarity, citation accuracy, and analytical depth. I provided line-by-line comments and suggestions below:
- Line 71–73: Although the authors have cited the review by Laaraj et al., the discussion remains general. Please include details of the specific microbial species that show connection between fatty acid metabolism and microbial-derived androgens to disease aggressiveness and tumour growth. Author response: This section has now been expanded to highlight several key points noted within the Laaraj review.
- Line 149 and 190: Please add relevant reference. Author response: This is now completed
- Section 2.1: acnes is isolated from multiple cancer types, including PCa. The authors have briefly mentioned C. acnes phylotypes; however, this section could be strengthened by adding a concise table or short paragraph describing each phylotype's characteristics and relevance to prostate tissue, as different lineages exhibit distinct pathogenic potential. Author response: additional information about C. acnes phylotypes and their relation to different infections and disease is now included in both the manuscript (lines 296 to 302) and figure 3.
- Figure 2: In the figure legend, the descriptions in Figure 2 and its legend do not correspond accurately. The text labeled as Figure 2A in the legend appears to describe the image shown in Figure 2B (and vice versa). Please double-check the figure labeling and legend to ensure they match correctly. Author response: This is now completed
- Line 257: The current citation [69] does not directly support the statement that “prostate cancer is generally considered to be an immunologically cold cancer.” Please cite more appropriate literature describing the immune-cold nature of PCa. Author response: This is now completed
- Line 257: For the statement “presence of acnes, which can stimulate inflammation,” please add appropriate citation. Author response: This is now completed
- Line 258: For the statement “alters the tumour microenvironment,” please add appropriate citation. Author response: This is now completed
- Section 3.1: The statement refers to “recent years,” but one of the cited studies (reference 32, 2014) is relatively dated. The authors should consider adding a more recent reference (published after 2020) to reflect current understanding of acnes-induced inflammatory signaling and to support the phrase “in recent years.” Author response: This is now completed
- Line 272–277: The description of the acnes inoculation model (Shinohara et al. [75]) is accurate but:
- The placement of the citation at the beginning makes the paragraph slightly confusing.
- Sentences are too long. I suggest repositioning the citation and shortening the second sentence so it clearly supports the entire set of findings and would improve readability. Author response: These sentences have now been revised for clarity
- Line 290–306: In this paragraph, authors described the connection between acnes with other ovarian and gastric cancers. It would be better understood by readers if the authors clarified that these additional references are provided as comparative evidence from other cancer types, rather than as direct evidence for prostate cancer. Author response: This section has now been modified to indicate that other cancer types are being discussed and compared to findings within prostate cancer
- Line 310–316: There is some confusion in this paragraph regarding the citation placement and the logical connection between sentences. The authors cite Ashida et al. [12] at the end of line 312 and then discuss host cell invasion... but it is unclear whether these findings originate from Ashida et al. [12] or from other studies. If this is separate studies, please add appropriate citations; if not, please rephrase the sentence so the same citation supports both statements. Author response: This is now completed
- Line 419: The citation [104] is repeated twice in the same sentence. Please cite it only once to improve clarity. Author response: This is now completed
- Line 420–425: The sentence is too long. It should be divided into shorter and clearer parts. Please rephrase it. Author response: This is now completed
- Line 478: “More recent work.” Please add reference(s) at the end of the sentence. Author response: The sentence beginning “most recent work” is the first sentence in a paragraph which discusses recent studies suggesting that C. acnes may be used therapeutically. Several studies are discussed in detail and cited throughout the paragraph. As such, including the references at the end of this sentence is redundant and repetitious.
- Line 479–483: The citation [123] is repeated twice in consecutive sentences. Please cite it only once to improve clarity. Author response: citation 123 now only appears once in the text
- Line 495: Change “1.15.1.” to “5.1.” Author response: This has now been corrected (line 512)
- Section 5.4: The in-text citation formatting appears inconsistent. The citation numbers (e.g., “[10, 107, 113]”) are larger than the surrounding text. Please ensure uniform font size and formatting across all references. Author response: This has now been corrected
- Section 5.5: Font size too large. Author response: The font size has now been changed to match the other subheadings
Reviewer 2 Report
Comments and Suggestions for Authors
1. These sections,1.1, 1.2 and 1.3 should be separate parts and should not be included in the introduction.
2. In Figure 2, IFN-y need to be revised, y to γ.
3. The titles in Part 5 are out of order and need to be corrected.
4. The conclusion part needs to be improved, this part does not have a good summary of the description of the whole text.
Author Response
- These sections,1, 1.2and 1.3 should be separate parts and should not be included in the introduction. Author response: The sections have now been clearly separated from the introduction and renumbers accordingly.
- In Figure 2, IFN-y need to be revised, y to γ. Author response: This has now been corrected
- The titles in Part 5 are out of order and need to be corrected. Author response: We are unsure which specific titles or ordering the reviewer was referring to, but we have reviewed and revised Part 5 to improve clarity and structure in line with the recommendations made by Reviewer 1 and 3. If there are particular sections the reviewer feels still require adjustment, we would be grateful for further guidance so we can address the concern more precisely.
- The conclusion part needs to be improved; this part does not have a good summary of the description of the whole text. Author response: the conclusion has been rewritten to provide a better summery of the review and key future research directions for the field.
Reviewer 3 Report
Comments and Suggestions for Authors
The authors present a timely and intriguing review on the potential role of Cutibacterium acnes as an emerging pathogen in prostate cancer. This is a topic of significant and growing importance in the field of cancer microbiology. The manuscript demonstrates considerable potential and provides a valuable compilation of existing evidence. However, it requires major revisions to enhance its scholarly rigor, structure, and visual appeal before it can be considered for publication.
- Abstract:
Upon its first mention in the abstract, the full name Cutibacterium acnes should be followed by its abbreviation in parentheses: (C. acnes). Thereafter, the abbreviation C. acnes can be used throughout the manuscript. Please add a list of abbreviations at the beginning of the manuscript.
- Manuscript-Wide Formatting and Style:
- Italicization: Please ensure that all microbial genus and species names, including Cutibacterium acnes and any other bacterial names, are italicized consistently throughout the entire text, including in the keywords.
- Reference Formatting in Text: The font for the reference numbers within the main body of the text must be changed to Times New Roman to match the rest of the manuscript.
- Citations: When citing the work of a specific researcher (e.g., "Smith et al. demonstrated..."), it is mandatory to include the publication year (e.g., "Smith et al. (2023) demonstrated...").
- Reference List: Please ensure that all references in the list include a Digital Object Identifier (DOI). Furthermore, all genus and species names within the reference list must be italicized.
- Introduction:
The introduction should be strengthened with a more substantial discussion on the critical importance of the early detection of prostate cancer. Please incorporate recent references to support this section.
The current introduction lacks depth regarding the role of pathogens in cancer development. Please add at least 2-3 new paragraphs discussing the established and emerging mechanisms by which various pathogens (bacterial and viral) contribute to oncogenesis. This will provide a broader context before focusing on C. acnes. Use recent, high-impact reviews or primary research articles as references.
Suggested references:
Lotfalizadeh N, Sadr S, Morovati S, Lotfalizadeh M, Hajjafari A, Borji H. A potential cure for tumor‐associated immunosuppression by Toxoplasma gondii. Cancer Reports. 2024 Feb;7(2):e1963.
Sadr S, Ghiassi S, Lotfalizadeh N, Simab PA, Hajjafari A, Borji H. Antitumor mechanisms of molecules secreted by Trypanosoma cruzi in colon and breast cancer: A review. Anti-Cancer Agents in Medicinal Chemistry-Anti-Cancer Agents). 2023 Sep 1;23(15):1710-21.
To provide a balanced perspective, the introduction should also discuss the potential therapeutic applications of pathogens or their components in oncology (e.g., bacterial-mediated cancer therapy). Please add at least 2-3 paragraphs on this topic, also supported by recent literature.
- Section Headings and Structure:
All subsection titles are currently too lengthy. They should be concise yet comprehensive enough to convey the section's content. Avoid using abbreviations in section headings.
Section 5: The title of this section should be changed to the more standard and concise "Future Prospects."
Section 5.5: The formatting of this sub-section's title is inconsistent with the rest of the manuscript. Please revise it to maintain a uniform structure throughout.
- Figures and Tables:
A strong review article benefits from high-quality visual summaries. The current number of figures and tables is insufficient (Only 2 figures). Please consider adding more schematic diagrams, tables summarizing comparative studies, or illustrative figures that depict the proposed mechanisms of C. acnes in prostate carcinogenesis. This will significantly improve the manuscript's readability and impact.
- Conclusion:
- The conclusion section should not contain references. It is meant to be a summary of your own findings and perspectives derived from the reviewed literature. Please remove the references currently cited in the conclusion.
Comments on the Quality of English Language
The English could be improved to more clearly express the research.
Author Response
Reviewer 3 comments
The authors present a timely and intriguing review on the potential role of Cutibacterium acnes as an emerging pathogen in prostate cancer. This is a topic of significant and growing importance in the field of cancer microbiology. The manuscript demonstrates considerable potential and provides a valuable compilation of existing evidence. However, it requires major revisions to enhance its scholarly rigor, structure, and visual appeal before it can be considered for publication.
- Abstract: Upon its first mention in the abstract, the full name Cutibacterium acnesshould be followed by its abbreviation in parentheses: ( acnes). Thereafter, the abbreviation C. acnes can be used throughout the manuscript. Author response: The abbreviation has now been added.
- Please add a list of abbreviations at the beginning of the manuscript. Author response: a list of abbreviations has now been added at the end of the manuscript (line 1140 onwards)
- Italicization:Please ensure that all microbial genus and species names, including Cutibacterium acnes and any other bacterial names, are italicized consistently throughout the entire text, including in the keywords. Author response: This has now been corrected
- Reference Formatting in Text:The font for the reference numbers within the main body of the text must be changed to Times New Roman to match the rest of the manuscript. Author response: This has now been corrected
- Citations:When citing the work of a specific researcher (e.g., "Smith et al. demonstrated..."), it is mandatory to include the publication year (e.g., "Smith et al. (2023) demonstrated..."). Author response: this is now corrected throughout the text
- Reference List:Please ensure that all references in the list include a Digital Object Identifier (DOI). Author response: Ref 8 is a website so no DOI available for this reference. Publications from Ref 135 and 137 were not provided with a DOI due to thier age and the publisher not including a DOI, but both references contain authors, title, journal name, and PMID numbers, allowing the article to be easily identified.
- Furthermore, all genus and species names within the reference list must be italicised. Author response: This has now been corrected
- Introduction: The introduction should be strengthened with a more substantial discussion on the critical importance of the early detection of prostate cancer. Please incorporate recent references to support this section. Author response: an additional section has now been added to the introduction describing the prostate structure, prostate cancer and its detection
- The current introduction lacks depth regarding the role of pathogens in cancer development. Please add at least 2-3 new paragraphs discussing the established and emerging mechanisms by which various pathogens (bacterial and viral) contribute to oncogenesis. This will provide a broader context before focusing on acnes. Use recent, high-impact reviews or primary research articles as references. Author response: A new section has now been added to the introduction, which discusses (in general terms) viral and bacterial contributors to oncogenesis. The reviewer recommended adding 2 to 3 paragraphs on this topic; however, only one additional paragraph has been added, as the subject is also addressed in more detail in subsequent sections with specific relevance to prostate cancer. A wider discussion of the oncogenic potential of microbes in general is outside the scope of this review.
- To provide a balanced perspective, the introduction should also discuss the potential therapeutic applications of pathogens or their components in oncology (e.g., bacterial-mediated cancer therapy). Please add at least 2-3 paragraphs on this topic, also supported by recent literature. Author response: A new section has now been added to the introduction, which notes the potential for using microbes as therapeutics. As the potential of C. acnes use in cancer therapy is discussed in greater detail in section 5.1 the added section is only 1 paragraph long (not 2-3 as the reviewer suggests).
- Section Headings and Structure: All subsection titles are currently too lengthy. They should be concise yet comprehensive enough to convey the section's content. Avoid using abbreviations in section headings. Author response: all subheadings have now been revised in line with the reviewers' suggestions and journal style.
- Section 5:The title of this section should be changed to the more standard and concise "Future Prospects.". Author response: This title has now been altered.
- Section 5.5:The formatting of this sub-section's title is inconsistent with the rest of the manuscript. Please revise it to maintain a uniform structure throughout. Author response: The size, font and spacing of the heading of section 5.5 is now consistent with other sub-section headings
- Figures and Tables: A strong review article benefits from high-quality visual summaries. The current number of figures and tables is insufficient (Only 2 figures). Please consider adding more schematic diagrams, tables summarizing comparative studies, or illustrative figures that depict the proposed mechanisms of acnesin prostate carcinogenesis. This will significantly improve the manuscript's readability and impact. Author response: In the initially submitted version of the manuscript, two figures and a summary table were included. Within the revised version of the manuscript we have also added an additional 2 figures to support the text.
- Conclusion: The conclusion section should not contain references. It is meant to be a summary of your own findings and perspectives derived from the reviewed literature. Please remove the references currently cited in the conclusion. Author response: This reference is now removed (line 626)
Round 2
Reviewer 1 Report
Comments and Suggestions for Authors
Line 161-165 and 168-170: The sentences are not clear. Please restructure them to improve clarity and readability.
Figure 1: The text within Figure 1 appears excessively large and is not consistent with the manuscript’s font style or size. Please adjust the figure text to ensure uniformity with the manuscript.
In Section 3.1: The table and two figures are placed consecutively without clear separation, which affects the readability of the section. Additionally, the figure legends are mismatched. Figure 2's legend corresponds to Figure 3 and vice versa. Please correct the labeling. The figures also appear uneven in size, with one figure noticeably larger than the other, please adjust them to ensure both figures are presented at a consistent and appropriate size.
Author Response
- Line 161-165 and 168-170: The sentences are not clear. Please restructure them to improve clarity and readability.
- Author response This section has now been edited for clarity
- Figure 1: The text within Figure 1 appears excessively large and is not consistent with the manuscript’s font style or size. Please adjust the figure text to ensure uniformity with the manuscript.
- Author response The text of figure 1 is now 12-point Calibri style, which is similar to the text of the submitted manuscript. A typo has also been corrected in this figure.
- In Section 3.1:The table and two figures are placed consecutively without clear separation, which affects the readability of the section. Additionally, the figure legends are mismatched. Figure 2's legend corresponds to Figure 3 and vice versa. Please correct the labeling. The figures also appear uneven in size, with one figure noticeably larger than the other, please adjust them to ensure both figures are presented at a consistent and appropriate size.
- Author response The table has now been moved to later in the text to provide better separation (now at Line 626). Figures 2 and 3 have been merged into a single figure and the in-text figure numbering and figure legends updated.
Reviewer 3 Report
Comments and Suggestions for Authors
Dear authors,
I am happy with the revision, excellent work.
MINOR COMMENTS:
Line 102: (Humphrey, 2017) must be numerical, not in APA7 format.
Still, the references are not according to the guidelines. The journal must be abbreviated.
Author Response
- Line 102: (Humphrey, 2017) must be numerical, not in APA7 format.
- Author response These in-text citations have been updated
- The references are not according to the guidelines. The journal must be abbreviated.
- Author response The journal names within the reference list have now been updated to abbreviated names